# Cytotoxicity and Genotoxicity of *Senecio vulgaris* L. Extracts: An In Vitro Assessment in HepG2 Liver Cells

**DOI:** 10.3390/ijerph192214824

**Published:** 2022-11-11

**Authors:** Mattia Acito, Carla Russo, Cristina Fatigoni, Federica Mercanti, Massimo Moretti, Milena Villarini

**Affiliations:** 1Department of Pharmaceutical Sciences, Unit of Public Health, University of Perugia, Via del Giochetto, 06122 Perugia, Italy; 2Sana Pianta Soc. Agricola S.a.s., Strada Tiberina Nord 228, 06134 Perugia, Italy; 3Inter-University Centre for the Environment (CIPLA-Centro Interuniversitario per l’Ambiente), University of Perugia, Piazza Università 1, 06123 Perugia, Italy

**Keywords:** *Senecio vulgaris*, herbal teas, pyrrolizidine alkaloids, genotoxicity, comet assay, safety assessment

## Abstract

*Senecio vulgaris* L. is a herbaceous species found worldwide. The demonstrated occurrence of pyrrolizidine alkaloids in this species and its ability to invade a great variety of habitats result in a serious risk of contamination of plant material batches addressed to the herbal teas market; this presents a potential health risk for consumers. In light of the above, this work aimed to assess the cytotoxic and genotoxic activity of *S. vulgaris* extracts in HepG2 cells. Dried plants were ground and extracted using two different methods, namely an organic solvent-based procedure (using methanol and chloroform), and an environmentally friendly extraction procedure (i.e., aqueous extraction), which mimicked the domestic preparation of herbal teas (5, 15, and 30 min of infusion). Extracts were then tested in HepG2 cells for their cytotoxic and genotoxic potentialities. Results were almost superimposable in both extracts, showing a slight loss in cell viability at the highest concentration tested, and a marked dose-dependent genotoxicity exerted by non-cytotoxic concentrations. It was found that the genotoxic effect is even more pronounced in aqueous extracts, which induced primary DNA damage after five minutes of infusion even at the lowest concentration tested. Given the broad intake of herbal infusions worldwide, this experimental approach might be proposed as a screening tool in the analysis of plant material lots addressed to the herbal infusion market.

## 1. Introduction

The genus *Senecio* (Asteraceae) is one of the largest genera of flowering plants and includes more than 1500 species [1]. The name for the genus *Senecio* is probably derived from “senex” (i.e., old man, in Latin), and this name was first used by Pliny in reference to the plant becoming grey and hairy when fruiting. The specific epithet *vulgaris* (i.e., common) refers to its wide distribution [2,3], and the binomial name *Senecio vulgaris* was first proposed by Carl von Linnaeus (1753) in his publication *Species Plantarum* [4].

*Senecio vulgaris* L. is a tetraploid, self-fertile, broadleaf annual herb (5–40 cm high), glabrous or slightly hairy, with pinnatifid or lobed leaves and green or purple stems [5]. It blooms from spring to autumn, giving yellow flower heads without ligulate flowers. Native to Mediterranean Europe, North Africa, and temperate Asia, *S. vulgaris* has now been spread worldwide by human activities and, because of its extensive ecological amplitude, can thrive in a variety of habitats—not only ruderal zones—including gardens, waste places, roadsides, and cultivated fields [6]. It is often known by the common names “groundsel” and “old-man-in-the-spring”; the name groundsel derives from the Anglo-Saxon “groundeswelge” literally meaning “ground swallower”, probably referring to the rapidity of weed spreading [7].

Several studies have presented the composition of *S. vulgaris* solvent extracts demonstrating the occurrence of bioactive compounds, such as flavonoids [8]. In the sixteenth and seventeenth centuries, *S. vulgaris* was commonly used for the treatment of a wide variety of health problems, including toothache, intestinal worms, amenorrhea, and dysmenorrhea [9]. Detrimental activities have also been reported. The ingestion of *S. vulgaris* and other species of this genus has been incriminated as a possible cause of hepatotoxicity in livestock: ingestion of floral and stem material by cattle and horses can lead to liver disease, resulting in weight loss, weakness, and death within nine months [5], with toxic effects that are usually latent until irreversible liver damage has occurred [10]. *S. vulgaris* biological features qualify it as a potentially very serious weed [11]. The intrinsic toxicity due to the production of hepatotoxic alkaloids makes the control of this species necessary in agricultural and pasture fields [5]. Exposure to *S. vulgaris* could also threaten human health through accidental poisoning due to ingestion of grain contaminated with weeds, consumption of herbal or bush teas, or when taken as herbal teas for medicinal purposes [12].

Pyrrolizidine alkaloids (PAs) represent a class of naturally occurring secondary metabolites that are present in over 6000 different plant species—approximately 2% of all flowering plants [13]—grown worldwide [14,15]. To date, more than 660 PAs and PA *N*-oxides have been identified [16,17]. Plants containing PAs primarily belong to the families of Asteraceae (alternate name: Compositae) (e.g., genus *Senecio*, *Eupatoria*), Boraginaceae (e.g., genus *Heliotropium*), and Fabaceae (alternate name: Leguminosae) (e.g., genus *Crotalaria*) [18]. In plants, PAs play a pivotal role in chemical defense against phytophagous insects [19]. After ingested by herbivores, PAs have been reported to be acutely toxic, genotoxic, and teratogenic to vertebrates and invertebrates, and chronic ingestion of PA-containing plants and contaminated hay, straw, or silage, has been reported to cause livestock poisoning [20,21,22,23]. In humans, exposure to PAs contaminating herbal teas, cereals, pollen, and honey has been related to acute and chronic hepatic toxicity [16,23,24,25], such as veno-occlusive disease, especially in children [26,27]. More recently, the presence of PAs has been reported in herbal and food supplements with a resulting exposure to these genotoxic/carcinogenic xenobiotics, without proper knowledge on the potential risks for human health consequent to this kind of exposure [28,29,30].

The qualitative and quantitative profiles of PAs in *S. vulgaris*—one of the most common species containing PAs worldwide—are influenced by several factors, including the geographical area of growth, season, and ontogenesis. A previous work investigating 12 different *S. vulgaris* populations in Europe and China [31] has detected 22 different PAs, including eight presumed PA *N*-oxides with unknown identities. In particular, senecionine, integerrimine, seneciphylline, and their *N*-oxides were present in the aerial parts of all plants and all populations. Spartioidine, retrorsine, and their *N*-oxides were found in all populations and in more than 90% of the individual shoot samples, whereas riddelliine *N*-oxide was detected in 10 populations. Senecivernine, usaramine *N*-oxide, and riddelliine were also detected. Several studies have investigated the occurrence of PAs according to the phenological development stages of plants and seasons, showing that during the year, the total PA concentration in *S. vulgaris* ranged from 1654.3 µg/g (growth stage five—inflorescence emergence, autumn) to 4910.2 µg/g (growth stage two—formation of side shoots, midsummer) [32,33]. Since some of these compounds have been shown to exert various toxic activities and have been included by the European Food Safety Authority (EFSA) Panel on contaminants in the food chain in the list of PAs to be monitored in food (i.e., senecionine, senecionine-*N*-oxide, senecivernine, seneciphylline, seneciphylline-*N*-oxide, retrorsine, retrorsine-*N*-oxide and, in addition, integerrimine and integerrimine-*N*-oxide) [17], the scenario of accidental contamination/ingestion of this plant should be taken into consideration.

In light of the above, this work aimed to investigate the cytotoxic and genotoxic activity of *S. vulgaris* chloroform and aqueous extracts in an in vitro liver preclinical model (i.e., HepG2 cells).

## 2. Materials and Methods

### 2.1. Chemicals, Reagents, and Media

All reagents used were of analytical grade. Chloroform, ethanol, hydrochloric acid (HCl), methanol, ethylenediaminetetracetic acid disodium (Na_2_EDTA) and tetrasodium (Na_4_EDTA) salt, sodium chloride (NaCl), sodium hydroxide (NaOH), and zinc were purchased from Carlo Erba Reagenti Srl (Milan, Italy). Dimethyl sulfoxide (DMSO), ethidium bromide, low- and normal-melting-point agarose (LMPA and NMPA, respectively), 4-nitroquinoline *N*-oxide (4NQO), tris (hydroxymethyl)aminomethane (Tris-HCl), and Triton X-100 were obtained from Sigma-Aldrich Srl (Milan, Italy). Acridine orange (AO), 6,4′-diamidino-2-phenylindole (DAPI), and Via1-Cassette^TM^ were purchased from ChemoMetec A/S (Allerød, Denmark). Eagle’s Minimum Essential Medium (MEM), and Dulbecco’s phosphate-buffered saline, pH 7.4 (DPBS) were purchased from Invitrogen Srl (Milan, Italy). Antibiotics (penicillin and streptomycin), fetal bovine serum (FBS), MEM non-essential amino acids (NEAA), sodium pyruvate, and trypsin were acquired from Euroclone SpA (Milan, Italy). Conventional microscope slides and coverslips were supplied by Knittel-Glaser GmbH (Braunschweig, Germany). Distilled water (dH_2_O) was used throughout the experiments.

### 2.2. Plant Material

Fresh aerial parts (including stems, leaves, and flower heads) of *S. vulgaris* were harvested in the locality of Tavernacce (43°14′11.5″ N 12°24′30.9″ E), Municipality of Perugia, Umbria Region, Italy. The botanical identification and authentication of the plants were performed by Dr. Andrea Primavera, Agronomist, President of FIPPO (Federazione Italiana Produttori Piante Officinali/Italian Federation of Officinal Plant Producers).

### 2.3. Extraction Methods

#### 2.3.1. Chloroform Extract

Aerial parts of *S. vulgaris* were dried, cut, and ground (5000 rpm, 10 s, 3 times) to a fine powder using an IKA^®^ Tube Mill Control (IKA-Werke GmbH and Co. KG, Staufen, Germany); then, 2 g of the powdered material was extracted for 16 h in 20 mL methanol [34] with occasional stirring using a mechanical shaker. The liquid portion was filtered, and the solvent was removed at a controlled temperature (40 °C) using a low-pressure evaporator. The residue was taken up and extracted in 10 mL of 2 M HCl and 10 mL of chloroform in a separatory funnel by vigorous shaking. After equilibrating the phases, the funnels were allowed to stand until phase separation occurred. Zinc was then added to the acid solution to convert the PA *N*-oxides to the parent compounds [35]. The 2 M HCl phase was then made basic to ensure the alkaloids are regenerated and then are subsequently extracted with 10 mL chloroform. The chloroform fractions were pooled and evaporated in a rotary evaporator under controlled temperature (40 °C) and low pressure. The residue was finally resuspended in DMSO for testing in HepG2 cells.

#### 2.3.2. Aqueous Extract

As above, aerial parts of *S. vulgaris* were dried, cut, and ground to a fine powder; 2 g of the powdered material was extracted by suspension in 100 mL boiling dH_2_O and infused for 5, 15, and 30 min. The suspensions were centrifuged (2000 rpm, 5 min), the supernatants were filtered through Whatman No. 1 filter paper, and they were finally freeze-dried (using Christ^®^ ALPHA I/5, Martin Christ Gefriertrocknungsanlagen GmbH, Osterode am Harz, Germany). After freeze-drying, the residues were then resuspended in complete culture medium (see below).

### 2.4. Cell Line and Culture Conditions

The HepG2 cells originate from human cancer cells collected in the 1970s from the liver of a young Argentine with a diagnosis of hepatoblastoma [36,37,38]. Thanks to a series of practical advantages (e.g., almost unlimited life span, stable phenotype, high availability, easy handling, etc.), these cells are commonly employed for liver toxicity investigation [39]. Moreover, compared with the use of other cell lines which demand the addition of exogenous metabolic activation systems (such as S9 mix), it has been demonstrated that the use of this model is more adherent to in vivo conditions [40].

The HepG2 cell line (ATCC HB 8065) was purchased from Istituto Zooprofilattico Sperimentale della Lombardia e dell’Emilia Romagna “Bruno Ubertini” (Brescia, Italy). The cells were grown in 25 cm^2^ flasks in MEM supplemented with 10% (*v/v*) FBS, 1% NEAA, 1 mM sodium pyruvate, 100 U/mL penicillin, and 0.1 mg/mL streptomycin, at 37 °C in a humidified atmosphere containing 5% CO_2_. HepG2 cells were sub-cultured by dispersal with 0.05% trypsin in 0.02% Na_4_EDTA (contact time: 5 min) and replated at a 1:2 dilution, which maintained cells in the exponential growth phase. Before the treatment, sub-confluent HepG2 cultures were detached by trypsinization and were suspended in complete MEM culture medium.

### 2.5. Cell Count and Viability: AO/DAPI Double Staining

According to the OECD (Organisation for Economic Co-operation and Development) guidance document for genetic toxicology testing [41], the highest soluble concentration of the test extracts for in vitro testing when the composition of the test substance is not defined (e.g., substance of unknown or variable composition, complex reaction products, biological materials, environmental extracts, mixtures of incompletely known composition) should be at least 5 mg/mL to increase the concentration of each of the components.

We assessed the possible cytotoxicity of the *S. vulgaris* extracts by evaluating 5 scalar concentrations (i.e., 50, 25, 10, 5, and 1 mg/mL, corresponding to 200, 100, 40, 20 and 4 mg dry matter – d.m.). For the test, the cells (2.5 × 10^5^ per well) were transferred into 12-well culture plates (Becton Dickinson Italia SpA, Milan, Italy) in 1 mL volume, and maintained in culture for 24 h. Then, cells were treated with 1% Triton X-100 (positive control) or *S. vulgaris* extracts over a range of 5 concentrations. The cells were exposed for 4 h. The number of total and viable cells was then assessed by staining the cells with AO and DAPI fluorophores, with AO staining the entire population of cells and DAPI staining nonviable cells. For the analysis, cell suspensions were loaded into Via1-Cassette which were then placed in a NucleoCounter^®^ NC-3000TM (Chemometec, Allerød, Denmark), a fluorescence-based image cytometer, where cell count and viability were determined [42,43]. Total cell concentration in Via1-Cassette was shown on a personal computer using NucleoView software.

### 2.6. Genotoxicity Testing: Comet Assay

To avoid the possibility of false-positive results arising from DNA damage associated with cytotoxicity [44], 3 non-cytotoxic concentrations of the *S. vulgaris* extracts (i.e., 1, 5, and 25 mg/mL) were addressed to the comet assay [45]. The test was conducted under alkaline conditions (alkaline unwinding/alkaline electrophoresis, pH > 13) following the original 3-layer procedure [45] as described in detail elsewhere [42].

Briefly, 48 h before the treatment with *S. vulgaris* extracts, HepG2 cells were trypsinized and seeded (approximately 5 × 10^5^ cells/well) in 12-well plates. Cells were then treated for 4 h with the *S. vulgaris* extracts; cell subcultures were also treated with the model mutagen 4NQO (1 µM; positive control). At the end of treatments, the cells were detached by trypsinization and collected by centrifugation (70× *g*, 8 min, 4 °C). Cell pellets were gently resuspended in 300 µL of 0.7% LMPA (in Ca^2+^/Mg^2+^-free DPBS, *w/v*) maintained at 37 °C, rapidly layered onto a conventional microscope slide precoated with 1% NMPA (in Ca^2+^/Mg^2+^-free DPBS, *w/v*), and finally covered with a coverslip. After brief agarose solidification at 4 °C, coverslips were removed, and the embedded cells were protected with a top layer of 75 µL of 0.7% LMPA. To lyse the embedded cells and to permit DNA unfolding, the slides were then immersed in cold, freshly prepared lysing solution (2.5 M NaCl, 100 mM Na_2_EDTA, 10 mM Tris-HCl; pH 10; 1% Triton X-100 added just before use) and left to stand for 18 h at 4 °C. The slides were then placed in a horizontal electrophoresis box (HE99; Hoefer Scientific, Holliston, MA, USA) filled with a freshly prepared buffer (10 mM Na_4_EDTA, 300 mM NaOH; pH > 13). Prior to electrophoresis, the slides were left in the alkaline buffer for 20 min to allow DNA unwinding and expression of alkali-labile damage. Electrophoresis runs were then performed in an ice bath for 20 min by applying an electric field of 1 V/cm and adjusting the current to 300 mA (Power Supply E411; Consort, Turnhout, Belgium). The microgels were then neutralized with 0.4 M Tris-HCl buffer (pH 7.5), fixed 10 min in ethanol, allowed to air-dry, and stored in slide boxes at room temperature until analysis. All steps of the comet assay were conducted in yellow light to prevent the occurrence of additional DNA damage.

Immediately before scoring, the air-dried slides were stained with 65 µL of 20 µg/mL ethidium bromide and covered with a coverslip. The comets in each microgel were blindly analyzed at ×200 magnification with an epi-fluorescent microscope (BX41, Olympus Co., Tokyo, Japan) equipped with a high-sensitivity black and white charge-coupled device (CCD) camera (PE2020, Pulnix Europe Ltd., Basingstoke, UK) under a 100 W high-pressure mercury lamp (HSH-1030-L, Ushio Inc., Tokyo, Japan) using appropriate optical filters (excitation: 510–550 nm; emission: 590 nm). Microphotographs were elaborated by Comet Assay III software (Perceptive Instruments, Suffolk, UK). A total of 100 randomly selected comets (50 cells/replicate slides) were considered for each experimental point. The extent of primary DNA damage was quantified as the percent of fluorescence migrated in the comet tail (i.e., tail intensity) [46].

## 3. Results

### 3.1. Cell Count and Viability: AO/DAPI Double Staining

Results concerning cell count and the viability assay are summarized in Figure 1. In HepG2 cells, viability decreased in a concentration-dependent manner after the treatment with both extracts (i.e., chloroform and aqueous extract), with a more marked cytotoxic effect of the highest concentration of the aqueous extract (5-and 30-min infusion). The results of viability assays determined the choice of concentrations to be investigated afterward; the highest extract concentration (i.e., 200 mg d.m.) was not assayed in the comet assay.

### 3.2. Genotoxicity Testing: Comet Assay

Genotoxicity testing was performed using three concentrations that did not show cytotoxic effects in the AO/DAPI assay (i.e., 4, 20, and 100 mg d.m). Figure 2 illustrates the results of the comet assay. 4NQO, used as a positive control, induced a marked response (*p* < 0.05), thus indicating the sensitivity of the test.

Exposure of HepG2 cells to *S. vulgaris* extracts significantly increased the extent of DNA damage in a concentration-dependent manner. This is quite clear in both chloroform and aqueous extracts, and, in the latter, this phenomenon is independent of the time of infusion. The highest concentration (100 mg d.m.) exerted a significant genotoxic effect in each extract (compared with the negative control), whereas 20 mg d.m. did so only in the aqueous extracts. Aqueous extracts obtained after 5 and 30 min of infusion induced primary DNA damage at the lowest concentration tested (4 mg d.m.).

## 4. Discussion

In this work, we investigated the biological activity of *S. vulgaris* extracts in a preclinical hepatic model, namely HepG2 cells.

First, we adopted an extraction procedure using methanol and chloroform (+ HCl), which represent widely used solvents for the extraction of PAs from plant material [47]. Results showed a slight loss in cell viability at the highest concentration tested, and a marked dose-dependent genotoxicity exerted by non-cytotoxic concentrations. Results are in line with the scientific literature, as evidence showed a low/null cytotoxic effect of retrorsine, riddelliine, senecionine, and seneciphylline in this cell line [48], and remarkable genotoxic effects of several PAs in liver cell lines, primary hepatocytes and other liver models [49].

The ability of *S. vulgaris* to invade a wide variety of habitats worldwide makes the contamination of other plant materials addressed to the herbal teas market an actual risk for the consumer. Consequently, to investigate a potential risk occurring in everyday life—unintentional exposure to *S. vulgaris* mediated by herbal infusion intake—we also performed all experimental sets using aqueous extracts, reproducing the preparation of herbal teas with three different times of infusion (i.e., 5, 15, and 30 min). Results of viability and genotoxicity assays are almost superimposable with those obtained using the organic solvent-based procedure. Therefore, although the qualitative and quantitative details are unknown, it can be stated that the compounds responsible for toxic activity are present in the aqueous extracts, which mimicked common herbal infusions. The genotoxic effect is even more pronounced in aqueous extracts (e.g., 20 mg d.m.). Moreover, aqueous extract induced primary DNA damage in HepG2 cells after five minutes of infusion, even at the lowest concentration tested. In our work, wild *S. vulgaris* plants were harvested in late September at growth stage five—inflorescence emergence. As the combination of this growth stage and season has been shown to correspond to the lowest PA concentrations in this species [32], the toxic effects might realistically be even higher in the case of contamination occurring during the harvest in spring and summer. Our study had several strengths. Firstly, we set up an accessible extraction method (aqueous extraction followed by freeze-drying), which is notably fast, cheap, and environmentally friendly. Moreover, water extraction perfectly fits with the aim of this study, namely a safety assessment in the context of unintentional exposure mediated by herbal tea/supplement intake. Secondly, the choice of three times of infusion kept the experimental design close to everyday life conditions, simulating the cases of a short/medium infusion time (5 and 15 min, respectively), and the case of a long infusion time (30 min), which might occur when the consumer accidentally forgets to take away the teabag from the hot water. Thirdly, our comparative analysis in both the chloroform and the aqueous extracts allowed us to verify the appropriacy of a *green* procedure in this kind of investigation.

Our study also had some limitations. Qualitative and quantitative analysis (e.g., LC-MS/MS) of PAs contained in both the chloroform and the aqueous extracts has not been carried out. However, given the complexity of such analyses when dealing with plant matrices, this kind of investigation deserves to be addressed in separate research.

## 5. Conclusions

In summary, through the use of an environmentally friendly extraction procedure, this work highlighted the in vitro hepatotoxic effect of *S. vulgaris* when considered in the form of herbal tea. Given the wide diffusion of this weed, the broad intake of herbal teas worldwide, and the clear genotoxic activity of extracts obtained by mimicking domestic tea preparation, this experimental approach might be proposed as a screening tool in the analysis of plant material lots addressed to the herbal infusion markets.

## Figures and Tables

**Figure 1 ijerph-19-14824-f001:**
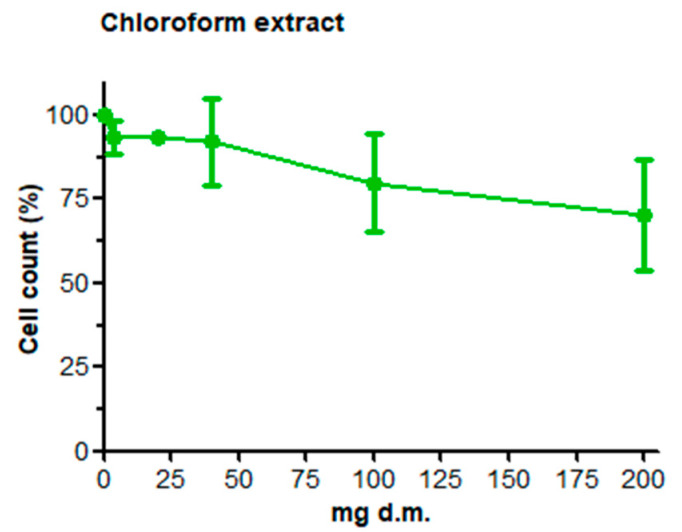
Effects of scalar concentrations of *S. vulgaris* chloroform and aqueous extracts on HepG2 cell viability expressed as percent of negative control (taken as the unit, 100%). The results of each experimental set are summarized as the mean (±SEM) of three independent experiments. Statistical analysis: * indicates statistically significant differences (*p* < 0.05) compared with the negative control, one-way ANOVA followed by Dunnett’s post hoc analysis. Cell viability (%) after treatment with the positive control (1% Triton-X-100): 34.94 ± 2.24.

**Figure 2 ijerph-19-14824-f002:**
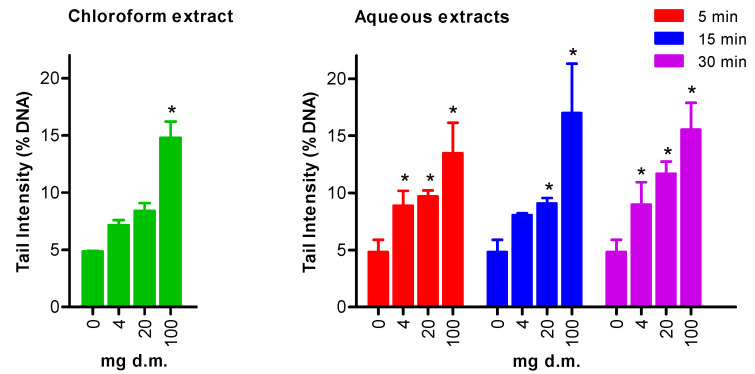
Primary DNA damage in HepG2 cells exposed for four hours to different concentrations of *Senecium vulgaris* chloroform and aqueous extracts. The extent of DNA strand breakage is expressed in terms of tail intensity (% DNA migrated in the comet tail). The results of each experimental set are summarized as the mean value of at least three independent experiments (±SEM). Statistical analysis: * indicates statistically significant differences (*p* < 0.05) compared with the negative control, one-way ANOVA followed by Dunnett’s post hoc analysis. Tail intensity of positive control (1 μM 4NQO): 18.55 ± 2.30.

## Data Availability

The data presented in this study are available within the article.

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
