# Peer review of "Cytotoxicity and Genotoxicity of Senecio vulgaris L. Extracts: An In Vitro Assessment in HepG2 Liver Cells"

_ijerph, 2022, doi:10.3390/ijerph192214824_

Round 1

Reviewer 1 Report

The authors present an interesting in vitro approach in researching the toxicity of Senecio vulgaris extracts on a hepatic cell line, the liver being defined as the organ most sensitive to the alkaloids present in this common plant.

Since this study concerns the genotoxic effect of S. vulgaris extracts, the authors could recall in the introduction which PAs are in majority present in the plant at the time of its harvest.

Line 134: there is twice "Moreover" at the beginning of the sentence.

To verify the potential effects of contamination of teas with S. vulgaris, did the authors plan to carry out a search for genotoxic effects by an experiment of measured additions of Senecio powder in a dose of tea before extraction with hot water for the proposed times followed by a test on HepG2 cells. This would make it possible to verify whether the molecules extracted from tea have an agonist or antagonist effect on the genotoxic efficacy of PAs.  

Author Response

The authors sincerely thank the Reviewer for constructive criticisms and valuable comments, which were of great help in revising the manuscript. Accordingly, the manuscript has been edited, as can be seen in the attached file with marked corrections.

Our responses to the Reviewer's comments are given in the attached file.

Reviewer 2 Report

In this study, the authors reported a preliminary toxicity assessment of the S. vulgaris extracts on the HepG2 cells. Both cytotoxic and genotoxic potencies were evaluated. Overall, this study is interesting and can be considered for publication after the following moderate revisions.

(1) Materials and methods: The purities of the chemicals and reagents need to be provided.

(2) In the discussion section, only two references are cited for discussions, which is not acceptable. It is strongly recommended to add more references and provide the in-depth discussions on the experimental results and mechanisms.

(3) How about the cytotoxicity and genotoxicity of the chloroform and methanol used for extraction?

(4) The research novelty and environmental implications should be highlighted in both the abstract and conclusion sections.

Author Response

(The authors gave the same response as above.)

Reviewer 3 Report

Very interesting study, but it is a pity that qualitative and quantitative analysis of PAs the extracts not defined. The solubility of substances, including pyrrolizidine alkaloids, largely depends on the nature of the solvent. It is logical that various compounds can be extracted with different solvents. In this regard, it is not entirely clear why the extracts obtained on the basis of organic solvents and water had similar cytotoxic and genotoxic effects?

It is desirable to present a drawing in introduction with basic toxic pyrrolizidine alkaloids which occur in like plants.

It would be possible to present the results in comparison with known cytotoxic and genotoxic compounds.

Line 122   instead of “two g” should be “2 g”

Line 134    delete extra «Moreover,».

Author Response

(The authors gave the same response as above.)

Reviewer 4 Report

Authors have investigated the cyto and geno-toxic potential of aqueous and organic extracts of Senecio vulgaris against HepG2 liver cell lines using well-known bioassays. The work is worthy of publication; however, I have few concerns as under:

1. At places methodological details are lacking/missing. For example, line 100, how much methanol was used? overnight (pl be specific, in hours)" under controlled temperature? (how? explain). How much Chloroform was used to extract each time?

Aqueous extracts: centrifugation details? 

2. What about the role of PA in inducing cyto and genotoxicity?

3. line 173: why treatment for 4 hour? any rationale?

line 179: dells?

line 185: details of buffer used.

Electrophoresis instrument model, make etc.

Another major concern is regarding the presentation. There is hardly any discussion. It is mostly the results which are explained without any interpretation using other published work. 

More so, the ms needs revision as regards the coherence and structuring; e.g. authors should provide a rationale for the use of the plant keeping in mind its medicinal use. It would have been better had the PA been tested/evaluated parallel to extracts. 

At most places, the language and presentation require attention. 

Author Response

(The authors gave the same response as above.)

Round 2

Reviewer 2 Report

My comments have been well addressed, and it can be accepted for publication.